# OpenReview forum: "Pre-Trained Vision-Language Model Selection and Reuse for Downstream Tasks"
_ICLR.cc/2025/Conference — Submitted to ICLR 2025_

### Official Review · Reviewer_sTKA · 2024-10-30

**Soundness:** 3
**Presentation:** 3
**Contribution:** 3
**Rating:** 8
**Confidence:** 4

**Summary:**

The paper explores a practical VLM reuse problem and proposes Model Label Learning (MLL), which efficiently selects and reuses pre-trained Vision-Language Models (VLMs) for downstream tasks. The framework consists of three modules: model labeling, which assigns labels to VLMs based on their capabilities; model selection, which matches these labels to task requirements; and model reuse, which employs an ensemble of selected models. In addition, a large-scale benchmark, including 49 VLMs and 17 datasets, is introduced to evaluate MLL’s effectiveness, with experimental results showing promising scalability and effectiveness.

**Strengths:**

1. The problem explored in this work is practical and meaningful. The proposed MLL framework provides an efficient way to select and reuse VLMs by leveraging a semantic graph and task-specific labels.

2. The method demonstrates good scalability. The use of a semantic graph allows MLL to expand as new models or tasks are added, making it adaptable to diverse visual tasks.

3. The paper is well-organized and easy to follow.

**Weaknesses:**

1. Regarding the scalability of the constructed semantic graph, if new nodes are added to the graph, is it necessary to add images to the sampled dataset to represent these new nodes? Additionally, have the authors considered using different datasets as the sampled dataset? If so, would different datasets impact the final performance?

2. For each target dataset, the highest performance achieved by any model in the model hub should also be included as a baseline result. This would help evaluate the effectiveness of the proposed method in selecting models.

3. As K is a core hyperparameter, more experiments analyzing its impact should be included, as the paper currently only presents results for K=1 and K=3. A more comprehensive analysis of K, including performance and computational cost at various values, is suggested.

**Questions:**

In addition to the points listed in weakness, the VLMs in the current model hub are primarily designed for image classification tasks. Have the authors considered expanding the proposed pipeline to accommodate more complex tasks, such as segmentation?

---

> ### Author Response · Authors · 2024-11-22
> **Rebuttal by Authors**
>
> Dear Reviewer sTKA:
> Thank you for the valuable feedback on our paper. We appreciate the time and effort you have put into reviewing our work. We have carefully read your review and addressed your concerns as follows.
> - Q1: Is it necessary to add images to the sampled dataset to represent these new nodes?
> - A1: It is not always necessary to add new images when new nodes are introduced to the semantic graph. If new nodes correspond to previously unseen classes, it’s possible to integrate them into the existing graph based on semantic relationships, even without adding new images. The semantic graph is not solely reliant on visual data but also on the connections and interrelations between classes.
> - Q2: Would different datasets as sampled dataset impact performance?
> - A2: We have considered other datasets as sampled dataset. However, other datasets do not offer the same extensive and diverse range of high-quality, reliable images across numerous classes and domains as ImageNet and its associated datasets. Previous work[1] has similarly utilized the ImageNet dataset to demonstrate the capabilities of VLMs, highlighting its established role in VLM selection. Therefore, we believe that the selected sampled dataset will hinder the model's ability to generalize effectively reflected by the model's labels.
> - Q3: Provide the highest performance achieved by any model in the model hub.
> - A3: We appreciate the reviewer’s valuable suggestion to enhance the practicality of the evaluation protocol. We have incorporated the evaluation of optimal performance on downstream tasks, which requires assessing the entire test dataset in the target task to select the best model.
> The results presented below demonstrate that our proposal outperforms the optimal approach on several downstream tasks, further highlighting its effectiveness.
> | | INB | ModelGPT | Proposal ($k$=1) | Proposal ($k$=3) | Optimal |
> | :----------------------: | :--------------------------: | :--------------------------: | :--------------------------: | :--------------------------: | :-------------------------: |
> | CIFAR100          | 0.8599 | 0.8599   | 0.8773 | 0.8923 | 0.8747   |
> | Country211        | 0.3121 | 0.3121   | 0.3159 | 0.3238   | 0.3514   |
> | CLEVR-D | 0.1940  | 0.1940    | 0.2111 | 0.1573  | 0.2221   |
> | DTD   | 0.6787 | 0.6787   | 0.6910  | 0.7053 | 0.6787   |
> | DMLab | 0.1262 | 0.1262   | 0.1361 | 0.1171 | 0.2482   |
> | Flowers102 | 0.8761 | 0.8761   | 0.8914           | 0.872            | 0.8761   |
> | MNIST| 0.7956 | 0.5648   | 0.8210            | 0.8101           | 0.8359   |
> | OxfordPet| 0.9401 | 0.9401   | 0.9488           | 0.9428           | 0.9498   |
> | PCam |0.5332| 0.4990    |0.5334| 0.5003           | 0.5986   |
> | FER2013 | 0.2859 | 0.4014   | 0.3904           | 0.4933           | 0.4466   |
> | Food101  | 0.9553 | 0.9553   | 0.9576           | 0.9566           | 0.9553   |
> | GTSRB   | 0.5391 | 0.5391   | 0.5752           | 0.5636           | 0.5391   |
> | RESISC45   | 0.6139 | 0.6139   | 0.6437           | 0.6800             | 0.6901   |
> | Rendered SST2     | 0.5199 | 0.5800     | 0.5206       | 0.5233           | 0.6249   |
> | StanfordCars      | 0.9487 | 0.9487   | 0.9568       | 0.9541           |0.9487|
> | STL10      | 0.9889 | 0.9639   | 0.9878     | 0.9854|0.9969|
> | UCF101    | 0.7702 | 0.7702   |0.7961| 0.8092           |0.8262|
> | Avg.    | 0.6434 | 0.6367   | 0.6620|0.6639| 0.6861|
> - Q4: Provide a more comprehensive analysis of $k$.
> - A4: We vary different k from 1 to 9 to form the ensemble predictor and compare the time complexity between $k$=1 and other situations.
> |$k$|1|2|3|4|5|6|
> | -------------------------------------------|------|------|------|------|------ |------|
> |**Avg Performance on 18 Target Datasets**|0.6621|0.6637|0.6639|0.6571|0.6556|0.6594|
> |**Interface Time Cost Compared with $k$ = 1**|1.000|1.927|2.233|2.467|2.700|2.900|
>
>   The result show that the optimal strategy would likely be to use a relatively small number of models, as they offer good performance with significantly lower computational expense compared to larger ensembles.
> - Q5:  Can the proposed pipeline expand to accommodate more complex tasks?
> - A5: We have considered the potential for extending our proposal to support more complex tasks. We believe that with modifications to our semantic graph and reuse metrics, the framework could be adapted to handle other tasks by incorporating task-specific components. We will explore this extension in future work for more advanced vision tasks.
>
> [1] LOVM: Language-Only Vision Model Selection. NeurIPS 2023.

---

> ### Comment · Reviewer_sTKA · 2024-11-26
>
> Thank the authors for addressing my concerns and considering my suggestions. I have carefully read all the comments and the authors' responses. I think the problem addressed by this work is interesting and the proposed approach is novel. The proposed framework is able to efficiently and effectively utilize pre-trained VLMs without access to all the test data in the target task. Therefore, I'd like to raise my score.

---

> > ### Author Response · Authors · 2024-11-27
> >
> > Dear Reviewer sTKA,
> >
> > Thank you very much for your acknowledgment and efforts. We are delighted to know that our response has addressed your comments. If there are any additional questions regarding our paper, we would provide further clarifications to address them.
> >
> > Best regards,
> > Authors

---

### Official Review · Reviewer_eqsx · 2024-11-01

**Soundness:** 3
**Presentation:** 3
**Contribution:** 3
**Rating:** 6
**Confidence:** 5

**Summary:**

The paper introduces Model Label Learning (MLL), a new approach for selecting and repurposing Vision-Language Models (VLMs) for downstream tasks. It comprises three main components: model labeling to categorize VLMs by their expertise, model selection to align VLMs with task requirements, and model reuse to integrate chosen VLMs into an ensemble for task application.

**Strengths:**

**[New perspective]** This work focuses on the selection and reuse of pre-trained VLMs to better suit the need of specific downstream tasks, which is novel and practical.

**[Good presentation]** This paper is well-written, making it easy to follow.

**[Thorough evaluation]** Extensive experiments have been done to evaluate the effectiveness of the proposed strategy.

**Weaknesses:**

**[Need more explanation]**
- In Figure 1, the details of the evaluated VLMs are missed. Please add this information in the caption for better understanding.
- The paper missed the introduction of ImageNet Baseline (INB). Is the best-performing model on ImageNet, i.e., EVA02-E-14?

**[Could be improved]**
- In line 245, this work randomly selects images $X_v$ from sample datasets to serve as representations for each node. Is there a more elegant solution for this, e.g., using the mean of several samples from the same class?
- For model reuse, the work selects top-k models with a simple ensemble approach. It would be nice to discuss or compare more advanced ensemble approaches in VLMs, e.g., “Beyond Sole Strength: Customized Ensembles for Generalized Vision-Language Models, ICML 2024”.

**[Experiments]**
- In Table 1, both INB and ModelGPT use the best-performing single model alone for evaluation. It would be nice to leverage them to select more models with ensemble for prediction when comparing the proposed method with 3-model ensemble. For example, the authors can select top-3 models on ImageNet for INB and do similar things for ModelGPT. Including this comparison can enhance the understanding of the effectiveness of the proposed method.
- Since the proposed MLL introduces three procedures, each costing extra time, could the authors provide the additional time introduced? This could offer insights on the trade-off between performance and time.

**Questions:**

Please refer to the weakness section.

---

> ### Author Response · Authors · 2024-11-22
> **Rebuttal by Authors**
>
> Dear Reviewer eqsx:
> Thank you for the valuable feedback on our paper. We appreciate the time and effort you have put into reviewing our work. We have carefully read your review and addressed your concerns as follows.
>
> - Q1: Provide the details of the evaluated VLMs in Figure 1.
> - A1: The evaluated VLMs in Figure 1 are completely consistent with the VLMs in the model hub of experiments in Section 5.
> - Q2: Provide the introduction of ImageNet Baseline (INB).
> - A2: The ImageNet Baseline (INB) method selects the best-performing model on the ImageNet dataset and reuses it on the downstream task. We thank the reviewer for pointing out these weakness and we will add detailed explanation for better understanding in the revision.
> - Q3: Is there a more elegant solution for selecting images from sample datasets?
> - A3: Thank you for your valuable suggestion regarding the selection of images. The current use of random selection in our proposal is a simple and effective method that allows for diverse representations without incurring extra computational costs. Moreover, we recognize the potential advantages of using means or other aggregate ways for selecting images, and we will experiment with those approaches and enhance our paradigm by these approaches in future studies.
> - Q4: Is there a more advanced ensemble approach for model reuse?
> - A4: Thank you for your constructive advice for improving the model reuse procedure.
> Firstly, we would like to clarify that our work focuses on a novel paradigm for selecting and reusing VLMs across downstream tasks, rather than just ensemble methods. The paradigm supports the reuse of one or more models and can incorporate various strategies, such as prompt learning or other advanced ensemble approaches.
> Therefore, while advanced ensemble approaches could enhance the paradigm, they are not the primary focus of our work. Moreover, we will explore the integration of more complex and advanced ensemble methods and their impact on performance and reuse effectiveness for our proposal.
> - Q5: Provide INB and ModelGPT with top-3 model ensembles for comparison with the proposed 3-model ensemble.
> - A5: We thank the reviewer for the constructive advice to make the evaluation protocol more practical. We conducted primary experiments that we selected the top-$k$ models from the comparison methods during the reuse phase, combined them into an ensemble for prediction, and evaluated the average performance of the ensemble on the target tasks. The following results show that our proposal is effective when model ensembles are applied.
> | | INB($k$=3) | ModelGPT($k$=3) | Proposal($k$=3) |
> | :-----------------: | :------: | :--------: | :--------: |
> | CIFAR100          | 0.8977 | 0.8949   | 0.8923   |
> | Country211        | 0.3228 | 0.3243   | 0.3238   |
> | CLEVR-D           | 0.1248 | 0.0907   | 0.1573   |
> | DTD               | 0.6968 | 0.6957   | 0.7053   |
> | DMLab             | 0.1296 | 0.1338   | 0.1171   |
> | Flowers102        | 0.8873 | 0.8876   | 0.8720    |
> | MNIST             | 0.7847 | 0.7888   | 0.8101   |
> | OxfordPet         | 0.9433 | 0.9490   | 0.9428   |
> | PCam              | 0.5002 | 0.5002   | 0.5003   |
> | FER2013           | 0.2636 | 0.2653   | 0.4933   |
> | Food101           | 0.9606 | 0.9611   | 0.9566   |
> | GTSRB             | 0.5938 | 0.5980   | 0.5636   |
> | RESISC45          | 0.6615 | 0.6555   | 0.6800     |
> | Rendered SST2     | 0.5332 | 0.5299   | 0.5233   |
> | StanfordCars      | 0.9555 | 0.9573   | 0.9541   |
> | STL10             | 0.9949 | 0.9957   | 0.9854   |
> | UCF101            | 0.7966 | 0.7933   | 0.8092   |
> | Avg.              | 0.6498 | 0.6483   | 0.6639   |
>
> - Q6: Provide the additional time introduced in the proposal.
> - A6: As discussed in section 3, our proposal is both time- and data-efficient, it generates model labels once only when the pre-trained model is uploaded to the model hub. In the model selection and reuse phase, we only generate target task captions and obtain reuse metrics for selection through vector computations, which is lightweight and time-acceptable. At present, the time complexity of our proposal increases linearly with the number of models in the model hub, which will not become a heavy burden when the number of models is large.

---

### Official Review · Reviewer_NHL5 · 2024-11-03

**Soundness:** 1
**Presentation:** 2
**Contribution:** 2
**Rating:** 5
**Confidence:** 4

**Summary:**

Selecting the best-performing pre-trained Vision-Language Models (VLMs) for a specific downstream task is challenging since no single VLM can achieve promising performance on all downstream tasks, and evaluating all available VLMs is impossible due to time and data limitations.
To address this problem, this paper proposes a novel paradigm to select and reuse VLM for downstream tasks, called Model Label Learning (MLL).
The proposal contains three key modules: model labeling, which assigns labels to each VLM to describe their specialty and utility; model selection, which matches the requirements of the target task with model labels; and model reuse, which applies selected VLMs to the target task in an ensemble manner.
The proposal is highly computationally efficient and growable since the model labeling process is completed target task independent and the ability could grow with the number of candidate VLMs.

**Strengths:**

- The proposed method is easy to understand.

**Weaknesses:**

- The novelty of the proposed method is weak. The main contribution of this paper is the model selection when ensembling multiple VLMs. However, there is no discussion or experimental analysis of the selected models during this process. What models are selected will give the readers a hint about the proposed method's characteristics or advantages.
- The analysis in this paper is too simple. After the model selection, what models are selected? As the main contribution is the model selection, the authors should show the selected models to understand the proposed method's characteristics and advantages.
- As a design choice analysis, the authors only tried K values to be 1 and 3. Although finding the best hyper-parameter is essential, why didn’t the authors try other values for K? The number of selecting models K is more important than the size of the model hub.
- Other essential design choice analyses are also missing. For example, in Eqn 8, why did the authors give high loss weight to models with high entropy? Is it the best choice of the weight values? Also, in Eqn 7, how is the hyper-parameter alpha decided, and how does it affect the model's performance?
- More importantly, the comparison with recent models is missing. There are several ways to improve VLMs without training, at least with the improved prompt-based approaches [1,2]. The authors should show the advantages of ensembling the models instead of the existing ways of improving VLMs. Also, ensembling models increases the number of total parameters. The authors should analyze the efficiency of the model ensemble compared to the existing approaches.

[1] Visual Classification via Description from Large Language Models. ICLR 2023.

[2] What does a platypus look like? Generating customized prompts for zero-shot image classification. ICCV 2023.

**Questions:**

Please refer to the questions in the weakness.

---

> ### Author Response · Authors · 2024-11-22
> **Rebuttal By Authors**
>
> Dear Reviewer NHL5:
>
> Thank you for the valuable feedback on our paper. We appreciate the time and effort you have put into reviewing our work . We have carefully read your review and addressed your concerns as follows.
> - Q1: The novelty of the proposed method.
> - A1: Thanks for your review, but it seems that you misunderstood our contribution. We highlight our contribution as follows:
> Firstly,  in recent years, an extensive range of pre-trained VLMs has been developed. Through extensive experiments in Section 1, we demonstrate that the performance of these models varies significantly across different downstream tasks. This highlights the critical need for an efficient approach to select the most suitable model for a specific task. To the best of our knowledge, this important problem has rarely been studied.
> Secondly, we propose a novel paradigm called Model Label Learning for VLM selection. The paradigm is both time- and data-efficient, and highly scalable. We believe that it can improve the ability of the existing model hubs, like HuggingFace, to make users select and reuse VLM more easily and conveniently.
> Thirdly, since the problem is novel, we present a new benchmark containing diverse downstream tasks for evaluating the methods, and the experimental results demonstrate the effectiveness and scalability of our proposal.
> Therefore, our paper focuses on designing a novel paradigm for selecting and reusing VLM, not just an ensemble method. It needs to be clarified that the reuse procedure of our paradigm can reuse one or multiple VLMs, and any model reuse method for VLM can be adopted, such as zero-shot or prompt learning. Moreover, since you proposed, we will provide a more detailed analysis of selected models for each task and extend our benchmark to include additional complex tasks, further validating the robustness and generalizability of our proposal across a broader range of real-world applications.
>
> - Q2: Provide a more comprehensive analysis of $k$.
> - A2: We vary different $k$ from 1 to 9 to form the ensemble predictor and compare the time complexity between $k$=1 and other situations. The result show that the optimal strategy would likely be to use a relatively small number of models, as they offer good performance with significantly lower computational expense compared to larger ensembles.
> |$k$|1|2|3|4|5|6|
> | -------------------------------------------|------|------|------|------|------ |------|
> |**Avg Performance on 18 Target Datasets**|0.6620|0.6637|0.6639|0.6571|0.6556|0.6594|
> |**Interface Time Cost Compared with $k$ = 1**|1.000|1.927|2.233|2.467|2.700|2.900|
>
> - Q3: Why give high loss weight to models with high entropy in Eqn 8?
> - A3: In VLMs, particularly those based on CLIP, predictions frequently exhibit overconfidence [1]. This overconfidence can degrade the performance of ensemble predictions, especially when the model makes incorrect predictions, as it amplifies errors.
> To address this issue, we introduce probability entropy as a measure of uncertainty and assign lower weights to models with high confidence when they are overconfident. This strategy reduces the impact of overconfidence, potentially erroneous predictions on the final ensemble output, thereby enhancing the robustness and accuracy of the model's overall predictions.
>
>
> [1] ConvNet vs Transformer, Supervised vs CLIP: Beyond ImageNet Accuracy. ICML 2024.

---

> > ### Author Response · Authors · 2024-11-22
> > **Rebuttal By Authors**
> >
> > - Q4: How is the hyper-parameter $\alpha$ decided, and how does it affect the model's performance in Eqn 7?
> > - A4:  Firstly, we emphasize that we use the same $\alpha$ across all datasets. Regarding the strategy for selecting $\alpha$, our experiments revealed that our method's performance is fairly robust to different $\alpha$ values, so we chose an $\alpha$ that performs well on our benchmark. To provide relevant analysis on this point, we varied the $\alpha$ values and demonstrated our method's performance.
> >   |$\alpha$|0.5|0.6|0.7|0.8|0.9|INB| ModelGPT |
> >   | ----------- |------ |------ |------ |------|------|------ |--------|
> >   | Avg. Performance|0.6569|0.6571|0.6620|0.6600|0.6590|0.6434|0.6367|
> >
> >   The experimental results show that, for different values of $\alpha$, our approach consistently outperforms the compared methods, demonstrating the robustness of our proposal to the choice of $\alpha$.
> >
> > - Q5: Why not comparing with recent works of improving VLMs without training?
> > - A5: With regard to the comparison with prompt-based methods, it is important to clarify that in our experimental evaluations, we employed only the most fundamental prompt-based techniques to assess downstream task performance. This choice ensures a fair and straightforward evaluation of our proposed method.
> > Furthermore, we view our approach as complementary to prompt-based methods rather than conflicting with them. By combining the strengths of both approaches, it is possible to achieve enhanced outcomes in downstream tasks, leveraging the benefits of model selection alongside the adaptability of prompt engineering.
> > We believe that the integration of our methodology with prompt-based approaches represents a promising direction for future research, offering the potential for significant improvements in the effective deployment of pre-trained VLMs across diverse applications.

---

> ### Comment · Reviewer_NHL5 · 2024-11-26
> **Response to the author rebuttal**
>
> The authors' rebuttal addressed most of my concerns. Therefore, I will raise my score.

---

> > ### Author Response · Authors · 2024-11-27
> >
> > Dear Reviewer NHL5,
> >
> > Thank you for reviewing and recognizing our work. We are pleased that our response has successfully addressed your comments. If there are any remaining issues or questions regarding our paper, we would be glad to address them to clarify our contributions further.
> >
> > Best regards,
> > Authors

---

### Meta-Review · Area_Chair_EPZU · 2024-12-15

**Metareview:**

The paper introduces an MLL framework that efficiently selects and reuses VLMs by leveraging a semantic graph, providing scalability for diverse visual tasks. The motivation is good, and the method demonstrates good scalability, making it adaptable to new models and tasks. The paper is well-organized and easy to follow. However, the novelty of the approach is limited, with insufficient analysis on model selection and hyperparameter choices. There is a lack of comparison to recent models and an inadequate discussion on the performance and efficiency of the proposed method.

Despite the reviewers acknowledging the contribution of the paper, the novelty remains a significant concern. After carefully reviewing the paper and the rebuttal, the AC, considering the emphasis on originality of ICLR, decided to reject the paper.

**Additional Comments On Reviewer Discussion:**

Most technical and presentation issues were addressed, but the reviewer who leans towards rejection maintained the original score based on the judgment of the core idea of the proposed method.

---

### Decision · Program_Chairs · 2025-01-22

Reject